# Investigation of the monopole magneto-chemical potential in spin ices using capacitive torque magnetometry

Naween Anand[1,5], Kevin Barry[1,2,6], Jennifer N. Neu[1,2,7], David E. Graf [1], Qing Huang [3], Haidong Zhou [3], Theo Siegrist[1,4], Hitesh J. Changlani [1,2] & Christianne Beekman [1,2 ✉]

The single-ion anisotropy and magnetic interactions in spin-ice systems give rise to unusual non-collinear spin textures, such as Pauling states and magnetic monopoles. The effective spin correlation strength ($J_{eff}$) determines the relative energies of the different spin-ice states. With this work, we display the capability of capacitive torque magnetometry in characterizing the magneto-chemical potential associated with monopole formation. We build a magnetic phase diagram of $Ho_2Ti_2O_7$, and show that the magneto-chemical potential depends on the spin sublattice ($\alpha$ or $\beta$), i.e., the Pauling state, involved in the transition. Monte Carlo simulations using the dipolar-spin-ice Hamiltonian support our findings of a sublattice-dependent magneto-chemical potential, but the model underestimates the $J_{eff}$ for the $\beta$-sublattice. Additional simulations, including next-nearest neighbor interactions ($J_2$), show that long-range exchange terms in the Hamiltonian are needed to describe the measurements. This demonstrates that torque magnetometry provides a sensitive test for $J_{eff}$ and the spin-spin interactions that contribute to it.

[1] National High Magnetic Field Laboratory, Tallahassee, FL 32310, USA. [2] Florida State University, Department of Physics, Tallahassee, FL 32306, USA. [3] University of Tennessee, Department of Physics, Knoxville, TN 37996, USA. [4] Florida Agricultural and Mechanical University and Florida State University, College of Engineering, Tallahassee, FL 32310, USA. [5] Present address: Intel Corp., Hillsboro, OR 97124, USA. [6] Present address: Ateios Systems, Newberry, IN 47449, USA. [7] Present address: Oak Ridge National Laboratory, Nuclear Nonproliferation Division, Oak Ridge, TN 37831, USA. ✉email: beekman@magnet.fsu.edu

Geometrically frustrated systems have an inherent incompatibility between the lattice geometry and the magnetic interactions resulting in macroscopically degenerate ground-state manifolds[1–5]. The large magnetocrystalline anisotropy and magnetic interactions in these systems give rise to unusual non-collinear spin textures, such as a spin-ice state that hosts emergent quasiparticle excitations equivalent to magnetic monopoles[6–9]. As in ref. [10], we denote the two-in/two-out Pauling states with (2:2), the 3-in/1-out monopole states as (3:1), and the all in/all-out configurations as (4:0). The effective spin-pair coupling ($J_{eff}$) determines the energy per tetrahedron for each of these states; only considering nearest-neighbor exchange interactions, $2J_{1,eff}$ is required to trigger the (2:2) → (3:1) transition. Importantly, the value of $J_{eff}$ is altered if interactions beyond nearest neighbor (i.e., dipolar $D$ and 2nd and 3rd nearest-neighbor exchange $J_2$, $J_3$) are included (see Table 1 and Fig. 1a), as described in previously reported models[1,11–14]. Similar to applied biases controlling the electro-chemical potential of electrons in a material, an applied field lowers the chemical potential of specific configurations leading to magnetic transitions between various non-collinear spin textures depending on the field direction and strength[2,10,15–17] (see Figs. 1, 2).

Field-induced phase transitions in these systems have been studied by magnetometry, neutron scattering, ultrasound and dilatometry techniques, experimentally, or through numerical methods[2,12,15,18–23]. In this work, we employ capacitive torque magnetometry (CTM) to characterize the spin-ice system $Ho_2Ti_2O_7$ (HTO) and to measure the effective spin-pair correlation strength between field-decoupled spins and the mean field. Conventional torque magnetometry is traditionally used to identify magnetic easy axes within crystalline materials[24]. However, the large magnetocrystalline anisotropy[25] makes HTO an ideal test-bed to reveal the unique capabilities of CTM in probing magnetic interaction energies, rather than the crystal field. From field dependent torque data, we extract the difference in magneto-chemical potential (MCP) between the (2:2) and (3:1) states, i.e., the MCP of monopole creation. Note, in the field range of this study, these are transitions between ordered states thus they cannot be classified as Kasteleyn transitions[26,27].

A striking result is that the extracted MCP ($2J^\alpha_{eff}$ and $2J^\beta_{eff}$) associated with monopole formation is different, depending on whether the monopoles nucleate on the $\alpha$- or $\beta$-spin sublattices[21] (see Figs. 1b, 2). While this conclusion is supported by classical Monte Carlo (MC) simulations using the standard (dipolar spin ice) (DSI) model (including dipolar interaction $D$, see Table 1), a comparison with the data clearly reveals the shortcomings of this form of the DSI Hamiltonian. Addition of a next-nearest-neighbor exchange term ($J_2 \sim 0.35$ K) improves the correspondence between the simulated and measured torque data for the transition involving the $\alpha$-spins, providing an estimate for $J_2$. This term only marginally increases the stability (i.e., angular range) of

the $(2:2)_X$ state, thus, a good agreement between the simulated and measured torque curves is still lacking for this transition. Additional third-nearest-neighbor exchange terms, $J^a_3$ and $J^b_3$, are therefore required to fully describe the field-induced phase transitions in HTO. The idea of needing long-range exchange interactions for the complete description of spin ices is not new. Values for $Dy_2Ti_2O_7$ have been extracted via modeling of susceptibility and neutron data[11,12,14,28], but to the best of the authors' knowledge, no such modeling has been reported for HTO.

## Results

**Torque rotations in the (001) and (1$\bar{1}$0) planes.** Torque magnetometry measurements have been performed on crystallographically oriented HTO single crystals as a function of external field strength, field direction, and temperature. Figures 3a, b show the torque responses when the field is rotated within the (001) and the (1$\bar{1}$0) plane of the unit cell, respectively. The zero-field contribution has been subtracted for all curves to show the magnetic response of the system, which is characterized by multiple sharp turnovers and zero crossings, with intermediate sinusoidal responses that are directly related to different crystallographic axes of the spin-ice system (see Methods and Supplementary Note 1).

A phenomenological single-unit-cell model is used to map out the evolution of magnetic phases as a function of field orientation for a given field strength. In this model, the sinusoidal torque curves are generated by explicitly calculating the torque response for one 16-site cubic unit cell for the different spin textures shown in Fig. 2 (solid curves in Fig. 3c, d) and intermediate mixed textures (dotted curves in Fig. 3c, d). These curves provide an objective way to determine the half-way point of each of the transitions. Based on a comparison of the model curves and the data, the $(2:2)_0$ is the only stable phase in the (001) plane, except near the [110] and the [$\bar{1}$10] directions when all $\beta$-spins flip and

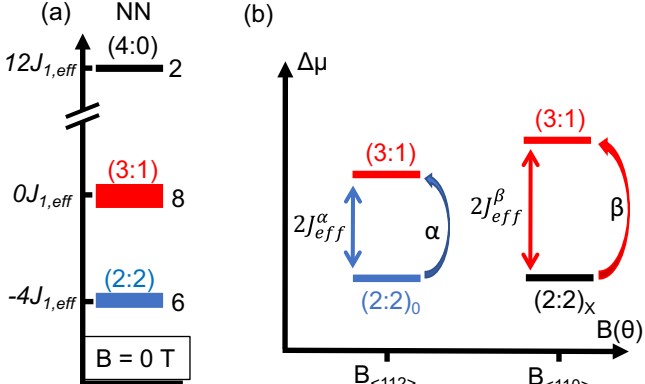

**Fig. 1 Magneto-chemical potential (MCP) of spin-ice states. a** Schematic of the total energy of the pyrochlore lattice per tetrahedron (NN model) for the spin-ice states at zero field (we assign $-J_{1,eff}$ to an in-out pair and $+J_{1,eff}$ to an in-in/out-out pair)[10]. Note, if an isolated tetrahedron is considered, the energy of the (2:2) states would be $-2J_{1,eff}$. Each tetrahedron could adopt one of six possible (2:2) Pauling states at low temperatures. Monopole states, i.e., a (3:1) tetrahedron (eight possible configurations), reside at higher energy and these states freeze out at low temperature. The all-in/all-out (4:0) (2 configurations) are at much higher energy. **b** Schematic of the chemical potential difference ($\Delta\mu$) between the Pauling states ($(2:2)_0$ and $(2:2)_X$) and the (3:1) state. The MCP, $2J^\alpha_{eff}$ and $2J^\beta_{eff}$, depends on the field direction, i.e., whether monopoles nucleate on the $\alpha$- or $\beta$-spin sublattice (defined in Fig. 2) for **B** ∥ ⟨112⟩ and **B** ∥ ⟨110⟩, respectively.

**Table 1 Interaction energy scales for HTO.**

| Model | Energy scales (K) |
|---|---|
| NN (no dipolar) | $J_1 = 5.40$, $J_{1,eff} = 1.8$ |
| Standard-DSI (NN-dipolar) | $J^{s-DSI}_1 = -1.56$, $D = 1.41$, $J^{s-DSI}_{1,eff} = \frac{J_1+5D}{3} = 1.83$ |
| Standard-DSI (long-range dipolar) | $J^{s-DSI}_1 = -1.56$, $D = 1.41$, $J^{s-DSI}_{1,eff} = \frac{J_1+4.53D}{3} = 1.61$ |
| Generalized-DSI (long-range exchange) | $J_1 = -1.56$, $D = 1.41$, $J_2 \sim 0.35$, $J_{2,eff} = \frac{J_2-D/\sqrt{3}}{3} = -0.155$, $J^a_3 = ?$, $J^b_3 = ?$ |

Values for the effective spin-pair interactions in HTO are provided for nearest-neighbor model (NN), the standard dipolar spin-ice model (s-DSI, NN-dipolar[13]), and for the s-DSI model with long-range dipolar interactions (i.e., Ewald Summation)[1,2,12]. The generalized DSI (g-DSI) model includes couplings up to third neighbors, which have also been shown to depend on phonon-induced distortions to the lattice[14]. In the g-DSI model, values have only been reported for $Dy_2Ti_2O_7$. The values presented in the bottom row are based on this work.

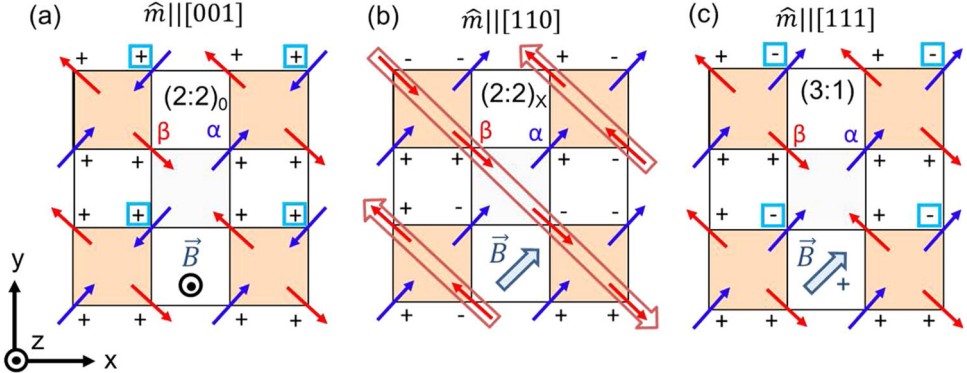

**Fig. 2 Spin textures of the ordered spin-ice states for different applied fields.** Spin-ice magnetic phases displayed as a 2D projection of the HTO unit cell down the $z$-axis. The orange squares are tetrahedra. The square in the center (gray) is not a tetrahedron; the diagonally opposing spins are in the same lattice plane. The $+/-$ signs indicate the spin directions along the $z$-axis[2,16,17,21]. **a** $(2{:}2)_0$ state with a net magnetic moment ($\hat{\mathbf{m}}$) in the $z$-direction ($(2{:}2)_0$ states also form with $\hat{\mathbf{m}}$ in the $x$- or $y$-directions, depending on the field direction); (**b**) $(2{:}2)_X$ state (with $\hat{\mathbf{m}} \parallel [110]$) in which the $\alpha$-spins (blue) are polarized and the $\beta$-spins (red) are antiferromagnetically aligned in chains (highlighted by the open arrows). The $(2{:}2)_X$ state also forms when **B** is directed along any of the family of $\langle 110 \rangle$ directions; and (**c**) $(3{:}1)$ state with one spin flip per tetrahedron with $\hat{\mathbf{m}} \parallel [111]$ (the $(3{:}1)$ state forms when **B** is directed along any of the family of $\langle 111 \rangle$ directions with $|B| \geq 2$ T). The spins denoted by the light blue boxes in panels (**a**) and (**c**) indicate the spin sublattice that becomes decoupled from the field when the magnetic field is directed exactly along the [112] direction. The direction of the magnetic field is indicated for each spin texture. The total energies per 16-site unit cell, i.e., interaction energies summed over 1st, 2nd and 3rd nearest neighbors and the Zeeman energies, are calculated for each of these ordered state, details regarding this calculation are provided in Supplementary Note 7.

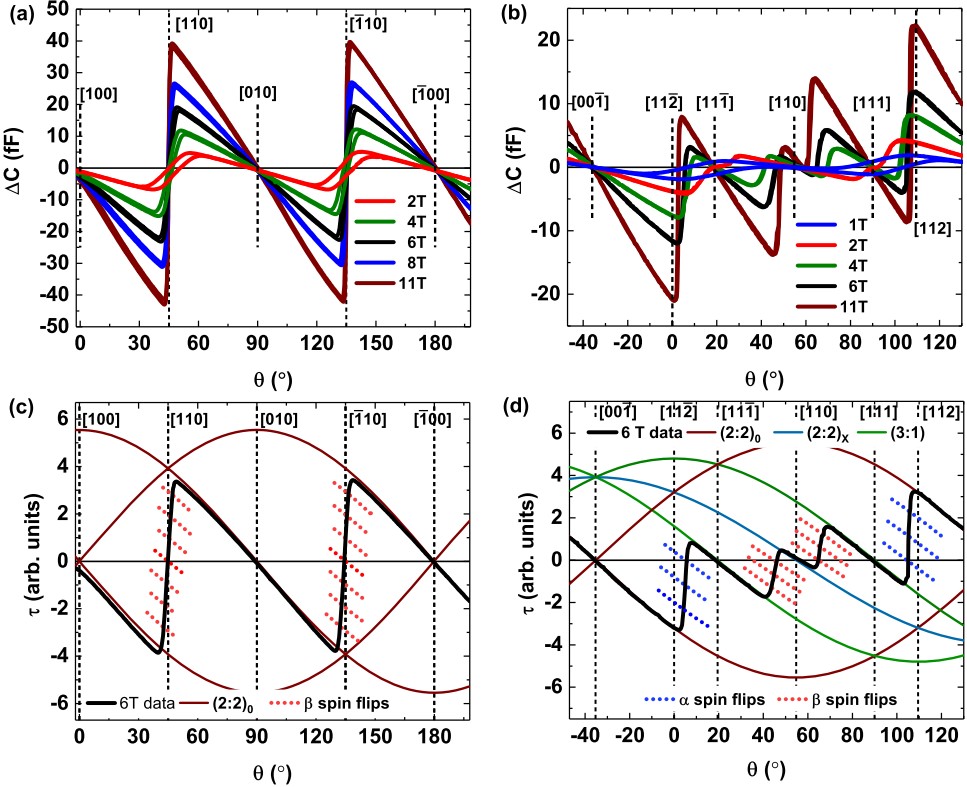

**Fig. 3 CTM angular measurements.** Measured capacitance change as a function of angle in various applied fields for an HTO single crystal, measured at $T = 0.5$ K. **a** CTM response when **B** rotates in the (001) plane (the [100] direction corresponds to 0°). **b** CTM response when **B** rotates in the ($1\bar{1}0$) plane (the [$11\bar{2}$] direction corresponds to 0°). **c**, **d** CTM response in a 6 T applied field for the (001) plane and the ($1\bar{1}0$) plane, respectively. The 6 T data (solid black line) was scaled according to sample volume and cantilever sensitivity. The data are plotted alongside calculated torque curves using the phenomenological model described in the Methods section and Supplementary Notes 1, 2. The solid colored model curves correspond to stable phases, while the dotted lines are calculated using volume fractions of spin flips on the $\alpha$ and $\beta$ sublattices. For each panel, crystallographic directions are indicated by vertical dashed lines.

the net magnetization sharply rotates by 90°. At low field, with the (3:1) states energetically out of reach, hysteresis appears around this transition, which is a clear sign of glassy behavior (see Fig. 3a and Supplementary Notes 1, 5).

For the rotation within the $(1\bar{1}0)$ plane all three spin textures show appreciable angular stability against misalignment of the field (see Fig. 3b, d). While this may not be surprising for the $(2:2)_0$ and (3:1) phases, we find the $(2:2)_X$ phase to be strikingly stable around the [110] direction, especially in small applied fields. Although a single-unit-cell model is not adequate to describe the long-range antiferromagnetic alignment of the $\beta$-spins of this phase, its stability indicates that a long-range ordered phase is present around this crystallographic direction, rather than a transient domain state as observed in the (001) plane rotation (Fig. 3a, c). [For field-angle phase diagrams, see Supplementary Fig. 2.]

A way to visualize the surprising anisotropy in the $(2:2)_X$ stability between the two rotation planes is to explore the energy surface that is obtained by integrating the torque curves. We show the energy surface contours associated with the (001) and the $(1\bar{1}0)$ rotation planes in Supplementary Fig. 3, with $(2:2)_X$ residing on a sharp maximum in the (001) plane and on a local minimum in the $(1\bar{1}0)$ plane. Thus, the $(2:2)_X$ phase resides on a saddle point in the energy landscape. While it is quite robust against misalignment of the field in the $(1\bar{1}0)$ plane, in the (001) plane the $(2:2)_X$ is not stable, and the system favors the $(2:2)_0$ states (i.e., a domain state with $\hat{\mathbf{m}} \parallel [100]$ and $\hat{\mathbf{m}} \parallel [010]$). The experimental observation of the $(2:2)_X$ phase is extremely sensitive to field misalignment, the high sensitivity of the CTM technique and the <1° accuracy of the polished crystal faces proved critical for our measurements.

Within the $(1\bar{1}0)$ plane, the $(2:2)_0 \Leftrightarrow (3:1)$ transition occurs when the field rotates across the [112] (and [11$\bar{2}$]) direction, when $\mathbf{B}$ is parallel to the [112] direction, one $\alpha$-spin per tetrahedron (Fig. 2a, c) becomes decoupled from the applied field[12,22,29]. The spins that are decoupled from the field maintain their spin-ice configuration due to the presence of the local internal field that is set by their spin environment. At a critical angle (i.e., a critical field) away from the [112] direction towards [111], the external field compensates the local internal field acting on the aforementioned spin sublattice allowing them to flip. From these critical angles at which $(2:2) \Leftrightarrow (3:1)$ transitions occur, we determine the MCP (i.e., the energy) associated with (3:1) state formation. Similarly, for the $(2:2)_X \Leftrightarrow (3:1)$ transition, when the applied field is aligned along the [110] (or equivalent) direction, there exist two $\beta$-spins per tetrahedron, which are decoupled from the field. The unit cell still maintains the spin-ice configuration, however that configuration is not unique, which leads to domains of degenerate magnetic phases. Theoretical and experimental evidence demonstrate the importance of second and third neighbor exchange couplings in addition to dipolar interactions[11,12,14,28], but evidence linking these correlations to the antiferromagnetic alignment of the decoupled $\beta$-chains is still lacking. In other words, these beyond-NN exchange interactions that are reportedly needed to stabilize the predicted low temperature ordered phase[14] involving alternating (single and double) spin chains, also play a role in stabilizing the $(2:2)_X$ phase at intermediate temperatures. We find ourselves well positioned to investigate the presence of these additional correlations because CTM allows us to extract the MCP of spin flip excitations for each of the sublattices separately.

**Monopole MCP extraction from CTM data**. The extracted critical angles are shown in the phase diagram in Fig. 4a for both transitions. By identifying the field-decoupled spin sublattice for

each transition, we fit the extracted angles as a function of applied field and extract the MCP associated with (3:1) monopole creation/annihilation. For the $(2:2)_0 \Leftrightarrow (3:1)$ phase transition, a value of $J_{eff}^{\alpha} = 1.61(5)$ K is determined from the experiment (details on the analysis are provided in the Methods section). In addition, if one extrapolates the fitted curves to the nearby $\langle 111 \rangle$ directions, a crossing point occurs at $B_c = 1.44$ T in each case. These crossing points match well with theoretical predictions[30,31], $B_c = 6J_{eff}^{\alpha}/(g\mu_B\langle J_z \rangle)$ and with experimental results ($B_m \approx 1.5$ T[23]) of the critical field required for the Kagome ice $\rightarrow$ (3:1) phase transition, which occurs as a function of increasing field when $\mathbf{B}$ is perfectly aligned along any of the $\langle 111 \rangle$ directions.

Strikingly, the same analysis for the $(2:2)_X \Leftrightarrow (3:1)$ transitions, yields a larger value of $J_{eff}^{\beta} = 2.2(1)$ K. We confirm this larger effective spin-pair coupling strength for the $(2:2)_X \Leftrightarrow (3:1)$ phase transition via field sweep measurements, with the field purposefully misaligned away from the [111] direction (see Fig. 4b). Surprisingly, the small misalignment of 5° away from the [111] direction (towards the [110] direction) stabilizes a low field $(2:2)_X$ phase (rather than a Kagome ice, expected when the field is perfectly aligned with any of the $\langle 111 \rangle$ directions), which transitions into the high field (3:1) monopole phase above a critical field[26]. We extract a critical field of 2 T for this transition, i.e., $J_{eff}^{\beta} = 2.1$ K, in line with the results from angular sweep torque data. While, the agreement between $J_{eff}^{\alpha} = 1.61(5)$ K and the predicted $J_{1,eff}^{s\text{-DSI}}$ with long-range dipolar interactions (see Table 1 and ref. [1]) is remarkable, the s-DSI model does not describe the $(2:2)_X \Leftrightarrow (3:1)$ transitions very well. As we will show below, the inclusion of higher order exchange terms affects the phase boundary and the stability of the spin-ice phases associated with both transitions.

Identical measurements were performed at $T = 1.7$ K, above the spin-freezing temperature[23] (see Supplementary Note 5). We find that beyond thermal smearing, the $(2:2)_X$ state is the only phase that changes significantly. This is evident from the change in slope of the torque curve around the [110] direction. This indicates deviation from a "clean" $(2:2)_X$ state due to thermal defects in the spin lattice at $T = 1.7$ K, which further supports the conclusion that the stable phase observed in CTM around the [110] direction is indeed the $(2:2)_X$ state.

**Monte Carlo simulated torque curves**. MC simulations were performed for a pyrochlore cluster with $16 \times 4^3 = 1024$ spins and periodic boundary conditions. Simulated torque curves are compared to the experiments. The results for a strictly nearest-neighbor model (NN, blue curve) and for the s-DSI model (including Ewald summation, loop moves, and demagnetization effects[2], red curve) are shown in Fig. 5a (see Supplementary Note 6 for more details). While the data are well described within either model at high field, it is clear that the experimental observations at low field are not fully described by either of these models. In low fields, the s-DSI model does well in approximating the critical angle associated with the transitions, but it over-estimates the angular stability of the (3:1) phase. In contrast, the NN model better approximates the (3:1) stability, but does less well with the critical angles. Most noticeable at higher fields, is that the stability of the $(2:2)_X$ phase is underestimated in both models.

We apply the same procedure for the extraction of the (3:1) MCP for each transition from the torque curves obtained from the MC simulations. The phase diagram based on the s-DSI model (with $J_2 = 0$) is presented in Fig. 5b. We obtain $J_{eff}^{\alpha,\text{MC}} = 1.4(2)$ K and $J_{eff}^{\beta,\text{MC}} = 1.8(1)$ K for the $(2:2)_0 \Leftrightarrow (3:1)$ and

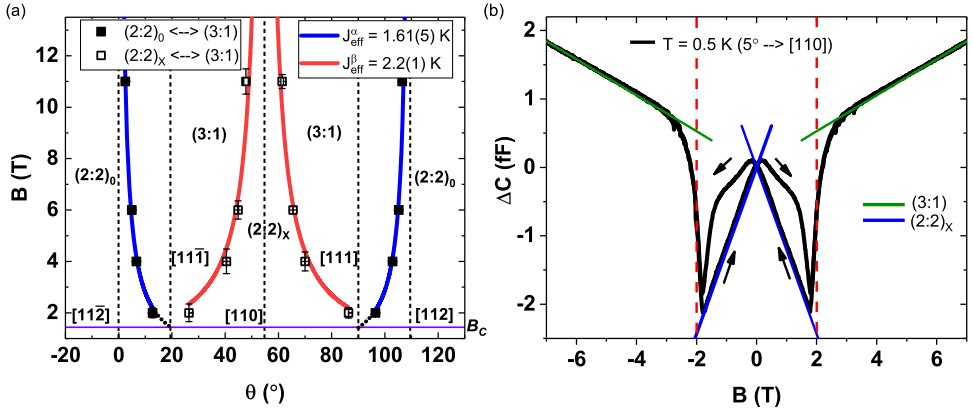

**Fig. 4 Monopole MCP extraction from experimental torque data. a** Critical angles extracted from the experimental data in Fig. 3, which define the transitions between the observed spin-ice magnetic phases, as a function of applied field (for the $(1\bar{1}0)$ plane rotations). The plotted error bars on the data points are a visualization of the residuals of the fits. Fits are shown for transitions between $(2:2)_0 \Leftrightarrow (3:1)$ (blue curves) as well as $(2:2)_X \Leftrightarrow (3:1)$ (red curves). The errors on the extracted $J_{eff}$ values are uncertainties obtained from the fits. The black dotted lines represent the extrapolation of the fit curves to the nearby $\langle 111 \rangle$ directions. The horizontal purple line represents the location of the critical field required to transition between the Kagome ice and the $(3:1)$ phases. Crystallographic directions are indicated by vertical dashed lines. **b** Capacitance change as a function of applied field at $T = 0.5$ K with **B** applied $\approx 5°$ away from the [111] direction, the black arrows indicate the sweep direction. The solid lines are volume-scaled calculated torque curves for the $(3:1)$ (green) and the $(2:2)_X$ phase (blue) (see Supplementary Note 4 for more details). The dashed vertical lines indicate the critical field of the $(2:2)_x$ to $(3:1)$ transition associated with a $J_{eff}^{\beta} = 2.1$ K.

$(2:2)_X \Leftrightarrow (3:1)$ transitions, respectively. The errors are based on the angular resolution (1°) of the simulations. While the qualitative trend is correct, these $J_{eff}$ values differ from results shown in Fig. 4a. The value for the $(2:2)_0 \Leftrightarrow (3:1)$ transition is only slightly smaller than the $J_{eff}^{\alpha}$ extracted from the measurements. That said, we note that there is a spread in reported values for the NN exchange and the dipolar interactions for spin-ice systems in existing literature[5,10,12,28], which could cause such a discrepancy. Similar to our experimental findings, the simulated curves show that the $(3:1)$ MCP is not the same, depending on the sublattice that the monopoles nucleate on during the transition. However, the $J_{eff}^{\beta,MC}$ extracted from the MC simulations for the $(2:2)_X \Leftrightarrow (3:1)$ transition is significantly smaller (1.8 K, Fig. 5b) compared to our experimentally observed value of $J_{eff}^{\beta} = 2.2$ K (see Fig. 4a).

In Fig. 5c, a snapshot of the spin texture in a $2 \times 2 \times 2$ unit cell structure is shown as a two-dimensional projection projected down the z-axis, illustrating the spin texture as extracted from the MC simulation at $T = 0.5$ K with $B = 4$ T $\parallel$[110] in the $(1\bar{1}0)$ plane. Under these conditions the ground state of the system is represented by a $(2:2)_X$ phase with no evidence of defects in the spin lattice. While the model does predict the correct ground state, it does not capture the entire extent of the angular stability of the $(2:2)_X$ phase.

To extend the DSI model beyond just the nearest-neighbor and dipolar terms, the minimal way is to add a next nearest-neighbor $J_2$ interaction. The presence of $J_2$ does not change the energetics of the $(3:1)$ phase, but for $J_2 > 0$ (see Methods) an additional Ising antiferromagnetic interaction is introduced. We have simulated curves for various $J_2$ values up to 0.04 meV (~0.464 K). In Fig. 5 we plot the simulated torque curve associated with the s-DSI model with $J_2 \sim 0.35$ K added to it. This term improves the agreement between the data and the MC simulations for the transition involving the $\alpha$-spins, now accurately approximating the $(3:1)$ stability at low fields, providing an estimate for the size of $J_2$ for HTO. The value of $|J_2/J_1|$ found in this work is similar to (but higher than) the reported value for the sister compound $Dy_2Ti_2O_7$[11,28]. However, while the angular stability of the $(2:2)_X$ phase did appear to marginally increase, the quantitative value of

the angular extent (see inset) is not explained by adding the $J_2$ term, indicating that interactions such as $J_3$, are necessary for a precise characterization of the Hamiltonian.

We support our findings with a short-range phenomenological model, which we use to evaluate the interaction energy for each spin-ice phase (see Supplementary Note 7 for more details). From this analysis, one can see what effect each of the interaction terms in the Hamiltonian has on the phase boundary of the field-induced magnetic phase transitions in HTO. In short, for the $(2:2)_0 \Leftrightarrow (3:1)$ transitions, the introduction of a $J_2$-term affects the interaction energy of the $(2:2)_0$ state, but does not impact the energetics of the $(3:1)$ state. Effectively, $J_2$ partially negates the effects of long-range dipolar interactions. Note, adding $J_3$-terms affects both the $(2:2)_0$ and $(3:1)$ states in the same way, thus this effect cancels out when evaluating the location of the phase boundary associated with this transition. (These $J_3$ terms correspond to two different kinds of third nearest neighbors, their couplings are referred to as $J_3^a$ and $J_3^b$, see Supplementary Note 7). For the $(2:2)_X \Leftrightarrow (3:1)$ transitions, the introduction of $J_2$ also does not affect the energetics of the $(2:2)_X$ phase, as the interaction energy associated with this term sums to zero (i.e., similar to the $(3:1)$ phase). Hence, the phase boundaries of the $(2:2)_X \Leftrightarrow (3:1)$ transitions are unaffected by the $J_2$ term, a finding broadly consistent with the MC simulations. However, the $J_3$ terms affect the $(2:2)_X$ and $(3:1)$ phases differently, and are therefore important in determining the location of the phase boundary for this transition. Thus, this simple short-range model allows us to constrain the value for $(J_3^a + J_3^b)$ to a ball-park value of $\sim -0.014$ meV ($-0.16$ K). The sign and the order of magnitude for $(J_3^a + J_3^b)$ are consistent with previously reported values for DTO[11].

While this work provides estimates for the interaction terms for HTO, owing to the strongly correlated nature of the system, a full re-optimization of all exchange parameters may be needed. An accurate determination of the individual values for $J_3^a$ and $J_3^b$ requires further extensive MC simulations, which we leave to future work.

In conclusion, we have shown that CTM can be used to evaluate the phase boundaries of magnetic phase transitions in

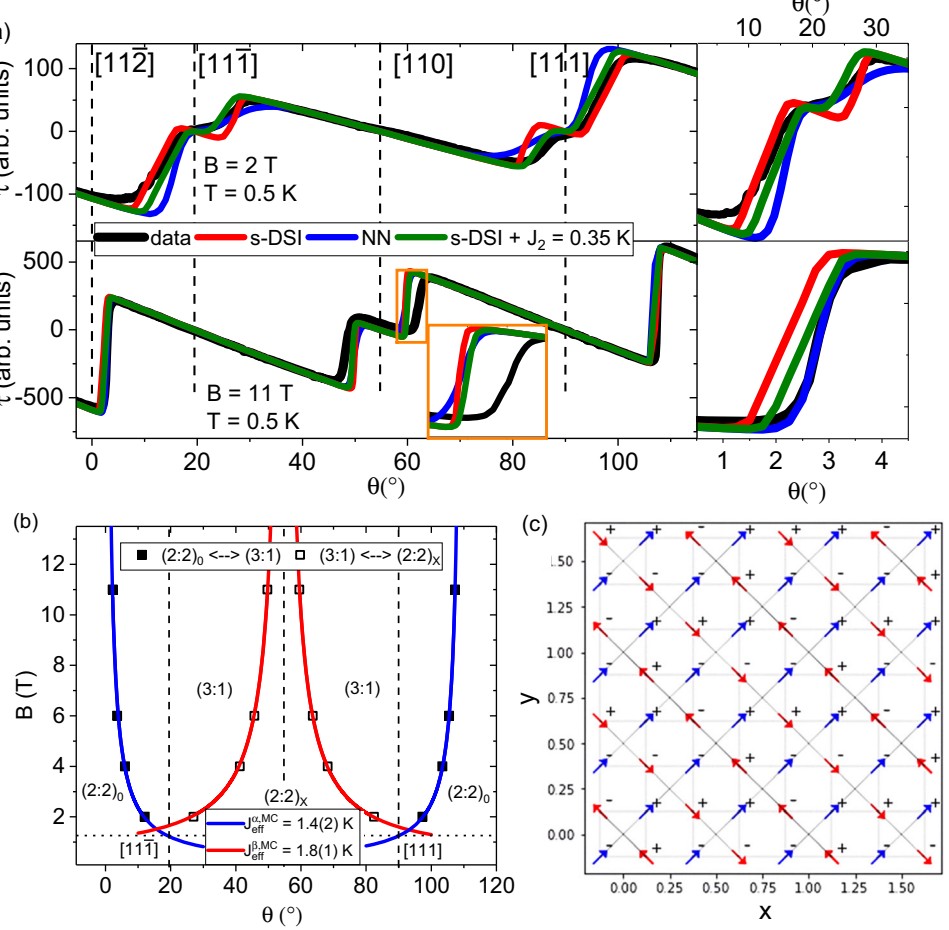

**Fig. 5 Monopole MCP extraction from MC-simulated torque curves. a** Simulated torque response as a function of angle, obtained from MC simulations using the nearest-neighbor model (NN, blue curves), s-DSI model (red curves) and the generalized DSI model (s-DSI with $J_2 = 0.35$ K, green curves); measured CTM data (black curves) are taken at $T = 0.5$ K in applied fields of 2 T (top panel) and 11 T (bottom panel). The inset highlights the angular region near the [110] direction (orange square). The panels on the right highlight the region around the [11$\bar{2}$] direction. Crystallographic directions are indicated with vertical dashed lines. **b** Critical angles extracted from the torque curves obtained from the MC simulations (s-DSI, with long-range dipolar $D$ and with $J_2 = 0$) for the transitions between the observed spin-ice magnetic phases, as a function of applied field ([1$\bar{1}$0] plane rotations). Error bars are based on the angular resolution (1°) of the simulations and are smaller than the symbol size. Fits are shown for transitions between $(2:2)_0 \Leftrightarrow (3:1)$ (blue curves) as well as $(2:2)_X \Leftrightarrow (3:1)$ (red curves). The uncertainties in the extracted $J_{eff}^{\alpha,MC}$ and $J_{eff}^{\beta,MC}$ values are errors determined from the fits. **c** A 2D snapshot of the spin texture (2x2x2 unit cells projected down the $z$-axis) taken during the MC simulation using $B = 4$ T, at $\theta = 50°$, and with $T = 0.5$ K. The blue and red arrows indicate the $\alpha$- and $\beta$-spin sublattices, respectively. The $+/-$signs indicate the spin directions along the $z$-axis. Under these conditions the ground state of the system is represented by a $(2:2)_X$ phase with no evidence of defects in the spin lattice.

spin-ice systems. The unique nature of the pyrochlore lattice and the spin-ice interactions allows us to evaluate the effects of $J_2$ and $J_3$ terms of the Hamiltonian separately, i.e., by investigating different phase transitions. We believe that CTM may serve as a natural complement to neutron scattering, specific heat, and magnetization measurements, which can be compared with careful numerics[32,33], as it can put stringent bounds on effective Hamiltonians and theories of magnetic materials, thereby aiding to complete the understanding of their low-energy properties and response to magnetic fields.

## Methods

**Single crystal growth.** Single crystal samples of HTO were grown using the optical floating-zone method. $Ho_2O_3$ and $TiO_2$ powders were mixed in a stoichiometric ratio and then annealed in air at 1450 °C for 40 h before growth in an optical zone furnace. The growth was achieved by zone melting with a pulling speed of 6 mm/h under 5 atm oxygen pressure. Single crystal x-ray diffraction experiments, taken on an Oxford-Diffraction Xcalibur-2 CCD diffractometer equipped with a graphite-monochromated MoK$_\alpha$ source, confirm the symmetry (Fd-3m) and lattice parameter of 10.0839(1) Å at 293 K, consistent with previous reports[3] (see Supplementary Note 8 for more details).

Crystallographic orientation and specific axis alignment was performed using an Enraf Nonius CAD4 4-circle single crystal x-ray diffractometer equipped with graphite-monochromated MoK$_\alpha$ radiation. Single crystals used for torque magnetometry measurements were prepared as cubes with 1 mm edge length. Crystallographic axis alignment to within 1° of the vector normal for each of the 6 polished faces was then confirmed, using single crystal x-ray diffraction, as a final check for each sample.

**Capacitive torque magnetometry.** Capacitive torque magnetometry measurements were performed at the National High Magnetic Field Laboratory in an 18 T vertical-bore superconducting magnet with a 3He insert allowing for an operating temperature range between 250 mK and 70 K. A calibrated Cernox resistance temperature sensor was used throughout our measurements to determine the sample temperature. Each single crystal sample was mounted onto a flexible BeCu cantilever, constituting the top plate of the parallel plate capacitor in our setup, and placed in an externally applied magnetic field while at low temperature (a schematic of the torque setup is provided in Supplementary Fig. 1). The applied magnetic field induces a torque, $\tau = \mathbf{m} \times \mathbf{B}$, on the magnetic sample causing the cantilever to deflect. This deflection yields a change in measured capacitance $\Delta C = C - C_0$ that is collected experimentally, where $C_0$ is the capacitance value collected in zero applied magnetic field. Here the magnitude of the induced torque $|\tau|$ is proportional to the change in capacitance ($|\tau| \propto \Delta C$) with a proportionality constant that is dictated by the elastic properties of the BeCu cantilever. An

Andeen-Harling AH2700A Capacitance Bridge operating at frequencies between 1000 and 7000 Hz was used to collect the capacitance data during each measurement. The measurement probe used allowed for rotation of the sample over a range of ~200° and a Hall Sensor was used to calibrate the sample rotation with respect to the applied magnetic field. Schematics of the (001) and $(1\bar{1}0)$ planes and the high symmetry axes that lie on these planes are provided in Supplementary Fig. 1.

**Phenomenological model.** In this study, we have employed a simple unit cell model to calculate the expected torque response as a function of angle for each of the stable spin textures (see Fig. 2) $(2:2)_0$, $(2:2)_X$ and $(3:1)$, and for the intermediate phases hosting an appropriate volume fraction of these spin textures. During each of the phase transitions, as the field is rotated within the $(1\bar{1}0)$ plane, the field-decoupled spins will flip to form intermediate domain states eventually leading to a $(3:1)$ monopole phase on all tetrahedra. Depending on the transition, these are either only $\alpha$ or only $\beta$ spins. When rotating within the (001) plane, the measured magnetic torque component is given as,

$$\tau_n = \hat{n} \cdot \tau = [0, 0, \bar{1}] \cdot (\mathbf{m} \times \mathbf{B}); \quad \mathbf{B} = B[\cos\theta, \sin\theta, 0] \quad (1)$$

When rotating within the $(1\bar{1}0)$ plane, these torque curves were calculated in the following way:

$$\tau_n = \hat{n} \cdot \tau = \frac{[\bar{1}, 1, 0]}{\sqrt{2}} \cdot (\mathbf{m} \times \mathbf{B}); \quad \mathbf{B} = B[\cos\theta \hat{j}' + \sin\theta \hat{k}'] \quad (2a)$$

$$\hat{i}' = \frac{[1, \bar{1}, 0]}{\sqrt{2}}; \quad \hat{j}' = \frac{[1, 1, \bar{2}]}{\sqrt{6}}; \quad \hat{k}' = \frac{[1, 1, 1]}{\sqrt{3}} \quad (\hat{i}' \times \hat{j}' = \hat{k}') \quad (2b)$$

Supplementary Tables 1 and 2 list all the moment vectors and the functional forms of the angular dependence of the torque curves for both rotational planes.

**Determination of $J_{eff}^{\alpha}$ and $J_{eff}^{\beta}$.** We examined the critical angles associated with each of the phase transitions observed in our torque vs. angle measurements. We define the critical angle to be marked by the location where half of all tetrahedra have a $(3:1)$ configuration. This angular position is extracted by finding the crossing point between the data and the associated model curve. Next, we identify which specific spin sublattice(s) decouple from the field and would be expected to flip when transitioning between the phases (see Figs. 1, 2). While one may expect that the $\alpha$- and $\beta$-spin sublattices decouple exactly at $\langle 112 \rangle$ and $\langle 110 \rangle$ field directions, respectively, the internal field produced by the mean field will shift that transition to a critical angle away from these crystallographic directions. Thus, the Zeeman energy ($E_Z$) associated with this critical angle is a direct measure of this internal field. We calculate the analytic form of the Zeeman energy of the field-decoupled spins as the field rotates across the [112] and [110] crystallographic directions, respectively. Expressing $E_Z$ in terms of applied field ($B$) and field direction ($\theta$), and realizing that $E_Z = 2J_{eff}$, allows us to determine a fitting function for the field vs. critical angle data from which the change in MCP ($J_{eff}^{\alpha}$ and $J_{eff}^{\beta}$) associated with the proliferation of $(3:1)$ tetrahedra can be determined. The results of the fitting are presented in Fig. 4a as the blue and red curves. For each type of spin sublattice, the MCP ($J_{eff}^{\alpha}$ and $J_{eff}^{\beta}$) takes the form

$$E_Z = \Delta\mu = -\mathbf{m} \cdot \mathbf{B} = -10\mu_B \hat{S} \cdot \mathbf{B} \quad (3)$$

where $\hat{S}$ represents the unit vector associated with the given spin sublattice of interest for the transition. This procedure allows the derivation of a functional form for $B(\theta)$, which is used to fit the extracted values for the critical angles as a function of external applied fields (see Fig. 4a). For the transition between the $(2:2)_0$ and $(3:1)$ phase near the [112] direction,

$$\Delta\mu = 2J_{eff}^{\alpha} = \frac{10\mu_B B}{\sqrt{18}} [4\cos(\theta) + \sqrt{2}\sin(\theta)] \quad (4)$$

The fitting function describing the transition near the symmetry-related $[11\bar{2}]$ direction, can be derived in the same way. For the transition between the $(2:2)_X$ and $(3:1)$ phase near the [110] direction,

$$\Delta\mu = 2J_{eff}^{\beta} = \frac{10\mu_B B}{\sqrt{18}} [-2\cos(\theta) + \sqrt{2}\sin(\theta)] \quad (5)$$

For the field sweep torque measurement in Fig. 4b, the applied field was misaligned by ~5° away from the [111] direction (towards the [110] direction), which stabilizes a low-field $(2:2)_X$ phase (rather than a Kagome ice, which is formed when the field is perfectly aligned with the $\langle 111 \rangle$ directions). The field sweep shows two markedly linear regimes when the field is swept from high field to zero. These linear regimes correspond to constant saturated magnetization values, the ratio between the slopes describing these linear regimes (high field: green curve; low field: blue curve) are in great agreement with the ratio of saturated magnetization expected for the $(3:1)$ (5 $\mu_B$/Ho) and $(2:2)_X$ (4.1 $\mu_B$/Ho) phase, respectively (see Supplementary Note 4). The field sweep also shows hysteresis around zero field, indicating a glassy response to a change in polarity of the applied field (i.e., the reversal of the $\alpha$ spins).

**Monte Carlo simulations.** We have simulated torque responses using the generalized DSI model whose Hamiltonian is given by

$$H = -J_1 \sum_{\langle i,j \rangle} \tilde{\mathbf{S}}_i \cdot \tilde{\mathbf{S}}_j - J_2 \sum_{\langle\langle i,j \rangle\rangle} \tilde{\mathbf{S}}_i \cdot \tilde{\mathbf{S}}_j + Dr_{nn}^3 \sum_{i>j} \left( \frac{\tilde{\mathbf{S}}_i \cdot \tilde{\mathbf{S}}_j}{|\mathbf{r}_{ij}|^3} - \frac{3(\tilde{\mathbf{S}}_i \cdot \mathbf{r}_{ij})(\tilde{\mathbf{S}}_j \cdot \mathbf{r}_{ij})}{|\mathbf{r}_{ij}|^5} \right) - g\mu_B \sum_i \mathbf{B} \cdot \tilde{\mathbf{S}}_i \quad (6)$$

where $\tilde{\mathbf{S}}_i$ are classical spin vectors with $|\tilde{\mathbf{S}}_i| = 1$. The tilde is used to indicate that the spins are constrained to point along the local $\langle 111 \rangle$ axis of the tetrahedra they belong to. $\mathbf{r}_i$ is the real-space location of site $i$, $\mathbf{r}_{ij} \equiv \mathbf{r}_i - \mathbf{r}_j$, $\langle i, j \rangle$ ($\langle\langle i, j \rangle\rangle$) refers to nearest-neighbor (next nearest-neighbor) bonds, $r_{nn}$ is the nearest-neighbor bond distance, $J_1$ ($J_2$) is the nearest-neighbor (next-nearest neighbor) interaction strength, and $D$ is the strength of the long-range dipolar term. The $i > j$ notation guarantees each of pair of spins is only counted once. $g\mu_B$ is the size of the magnetic moment and $\mathbf{B}$ is the applied magnetic field.

Our calculations were performed for finite-size pyrochlore clusters (16 atoms per simple-cubic unit cell) with $N_{spins} = 16 \times 4^3 = 1024$ lattice sites, and with periodic boundary conditions. For the nearest-neighbor model, we set $J_1 = +5.40$ K and $D = 0$. To deal with long-range magnetic dipolar interactions, the Ewald summation technique was employed to convert the real-space sum in the Hamiltonian into two rapidly convergent series, one in real space and the other in momentum space. This Hamiltonian was then simulated with the Metropolis Monte Carlo algorithm, using a combination of single spin flip and loop moves[2] (which allows a ring of spins to flip at one time, while maintaining the ice rule constraint). For the s-DSI simulation (with long-range dipolar interaction $D$, red curve in Fig. 5a), the parameters were set to $J_1 = -1.56$ K, $D = 1.41$ K and $g = 10$[5,10]. $J_2$ was varied to investigate its effect on the critical angles associated with both transitions. To produce the green curves in Fig. 5, $J_2 = 0.35$ K was used, additional simulations using different values for $J_2$ are presented in Supplementary Fig. 6. Demagnetization effects (assuming a spherical sample) were taken into account[2] for the presented simulated torque curves. More details of our simulations can be found in Supplementary Note 6.

## Data availability

The authors declare that the main data supporting the findings of this study are available within the paper and its Supplementary Information. The crystallographic data have been deposited with the joint CCDC/FIZ Karlsruhe online deposition service under no. CSD-2172269[34]. Other data that support the findings of this study are available from the corresponding author upon reasonable request.

## Code availability

The code used to generate the Monte Carlo simulation results shown in the paper is publicly available at https://github.com/hiteshjc/Ising_Ice_dipolar Additional scripts and files for the numerical calculations are available from H.J.C. upon reasonable request.

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

## Acknowledgements

C.B. and K.B. acknowledge support from the National Research Foundation, under grant NSF DMR-1847887. J.N. and T.S. acknowledge support from the National Research Foundation, under grant NSF DMR-1606952. A portion of this work was performed at the National High Magnetic Field Laboratory, which is supported by National Science Foundation Cooperative Agreement No. DMR-1157490, No. DMR-1644779, and the State of Florida. Q.H. acknowledges support from the National Research Foundation, under grant NSF-DMR-2003117. H.D.Z acknowledges support from the NHMFL Visiting Scientist Program, which is supported by NSF Cooperative Agreement No. DMR-1157490 and the State of Florida. H.J.C. acknowledges support from the National Research Foundation, under grant NSF DMR-2046570, and start-up funds from Florida State University and the National High Magnetic Field Laboratory. The simulations were performed on the Research Computing Cluster (RCC) and the Planck cluster at Florida State University. We thank R. Moessner and L. Jaubert for helpful discussions.

## Author contributions

C.B. conceived the experiment(s) and analyzed the results, N.A. and K.B. contributed equally to this work, they conducted the torque magnetometry measurements and analyzed the results, D.G. assisted in conducting the torque magnetometry measurements, Q.H. and H.Z. synthesized the single crystals, J.N. and T.S. oriented and polished the crystals. H.J.C. performed the Monte Carlo simulations and contributed to the theoretical analysis. All authors reviewed the manuscript.

## Competing interests

The authors declare no competing interests
