## [Peer Review File · Nature Communications]

REVIEWER COMMENTS

Reviewer #1 (Remarks to the Author):

This manuscript presents a study by torque magnetometry, augmented by modeling and Monte Carlo simulations, of the high-field phases of a classical spin ice material, $\text{Ho}_2\text{Ti}_2\text{O}_7$. The data shown are of a high quality, and are analyzed and discussed in a compelling way. The established phase diagram will certainly be of interest to the workers in the field, and overall the study clearly deserves dissemination.

However, in my opinion it is less clear whether Nature Communications or a more specialized journal is a more appropriate outlet. In both the abstract and the concluding paragraph, the authors stress the method of torque magnetometry – or more specifically capacitive torque magnetometry (CTM) as having a unique capability in investigating spin ices and other highly anisotropic systems. CTM is not a very new technique, and together with other types (e.g. piezoresistive) torque magnetometry has been long used to study anisotropic systems (such as anisotropic superconductors). With regard to establishing phase boundaries in spin ice materials (and to link those as well as additional magnetic information to model hamiltonians), other magnetic techniques seem to work as well (for example, ac susceptibility combined with a vector magnet, had been applied by Borzi et al., Nat. Commun. 7:12592 on $\text{Ho}_2\text{Ti}_2\text{O}_7$ and $\text{Dy}_2\text{Ti}_2\text{O}_7$ to that effect (although the region of fields and angles probed in that study was more limited).

Apart from the methodological aspect, the manuscript also presents conclusions about the physics of $\text{Ho}_2\text{Ti}_2\text{O}_7$. As mentioned in the last sentence of the abstract, the analysis with Monte Carlo simulations suggests that longer-range than nearest neighbor exchange interactions need to be included in the Hamiltonian for spin ices. This is convincingly demonstrated, although the paper could have gained by trying to estimate what those further terms should be to be consistent with the experimental results. From past publications (including the specifically cited refs. 16, 21, 22 and the paper by Borzi and coworkers) it seems to me that this was already pretty well established before.

In summary, while I do think this is a very nice paper, I do not see the proof of an important advance established by the manuscript, at least in its present form.

Reviewer #2 (Remarks to the Author):

The manuscript under consideration employs capacitive torque magnetometry technique to explore the magnetic phase diagram of the spin-ice material $\text{Ho}_2\text{Ti}_2\text{O}_7$. The authors focus on the higher-field region where they are able to extract the critical angles from their experimental data. Overall, the manuscript makes a good impression by providing new experimental data on a well-studied compound backed-up by the Monte Carlo simulations. Supplementary materials give detailed information on the geometry of the measurements, field sweeps etc. proving author's very careful approach to the data analysis. I am therefore prepared to recommend the manuscript for publication in Nature Communications.

The only comment I have is related to the hysteresis observed in lower fields. It has been shown from the low-temperature magnetisation measurements that field-sweeping rate is important around the spin freezing temperature (the relaxation times become very large). Could it be that the speed of changing the angle for applied field is also important in that respect and that if the direction is changed rather rapidly, the system is effectively out of the equilibrium?

Further comments:

Page 1, the authors claim that "... these systems have been mainly studied by magnetometry, neutron scattering, analytically, or through numerical methods" This is largely true, however, the dilatometry and ultrasound techniques have also been used for probing field-induced states in the spin ice compound, they should also be mentioned and the appropriate references given.

Fig. 2. The x-axis labelling for panel (a) should be made identical to the one used on panel (c), as the marks for 50, 100, 150 degrees are not particularly useful.

Fig. 3. A second caption above panel (a) is unnecessary. Same comment for Fig. 4(b).

Supplementary materials. Not sure the meaning of 'measurements' in the section 5 title, "Hamiltonian, Monte Carlo Simulations and Measurements".

Reviewer #3 (Remarks to the Author):

The manuscript presents a study of transitions between ordered states in spin ice compound $\text{Ho}_2\text{Ti}_2\text{O}_7$ under external magnetic field rotating in two planes. The capacitive torque magnetometry is used to detect spin flip transitions as a function of the strength of the external field and the field

direction. A model including nearest neighbor interaction and the Zeeman term is fitted to the experimental data and the effective nearest neighbor interaction constants are obtained.

In my opinion the manuscript is in many places unclear or inconsistent and it is difficult to assess the main results and the novelty of the study. Some additional explanations would be needed for the clarity of the paper, and there are also points that I do not understand. For example, the estimation of the effective interaction term J_{eff} is one of the key results of the study, however, J_{eff} is not properly defined or introduced at the beginning of the paper and it is used in a way that I find confusing. Several detailed comments are listed below.

1. I assume J_{eff} to be the effective nearest neighbor interaction between spins, defined as in the caption of Fig. 1 and leading to energy levels shown in the left panel of Fig 1a. However, I do not understand the notation in the right panel of Fig 1a (not adequately explained in the caption):

- the axis label " E_z ": E_z is defined later in the text as the Zeeman energy, however, there is also the interaction term leading to the energy levels plotted in the figure,

- for B [100], the splitting of the energy levels of (2:2) states should be 1:4:1, not 1:5. Also, the (3:1) states would split. The energy difference 3.2 K, if I understand correctly, comes from the spin-spin interaction term, so the gap between the lowest (2:2) state and the lowest (3:1) state should be modified,

- for B[110], the authors decided to show levels of only two (2:2) states that are the building blocks of the (2:2) \times configuration with larger unit cell. However, the (3:1) states should be split, if all 8 levels are shown.

- the $B > 2 T$ condition should be clarified.

2. J_{eff} is described in the text as field dependent, or field-direction dependent. After reading the whole text I suppose it means that (two) different values of J_{eff} were extracted from the data, depending on the magnetic field direction during the transition from one of (2:2) states to a (3:1) state. As discussed in the text, the different values might result from long range interactions present in the spin states involved in these transitions. Describing J_{eff} as field-dependent might be confusing.

3. A term "field decoupled spins" is used several times in the manuscript. Indeed, one or two spin sublattices become perpendicular to the field for some direction of B, for example B parallel to [112] or [110].

However, I do not understand some statements in the text:

- J_{eff} described as the interaction strength between field-decoupled spins (page 1),

- in Figure 1, the light blue box is described as indicating the spin sublattice that becomes field decoupled when the field is “tilted away from the [112] direction” (the spins are decoupled from field precisely for [112] direction, and become coupled to a field component as the field changes),

- “By identifying the field-decoupled spin sublattice for each transition, we fit the extracted angles against critical field and extract the MCP associated with (3:1) monopole creation/annihilation” [when B is rotated, the Zeeman energy of a given spin sublattice changes and this might lead to spin flips in this sublattice. However, not always spins decoupled for some field direction are flipped in the transition, for example only half of the beta spins, decoupled at B parallel to [110], flip in the transition from a (2:2)x to (3:1) state].

4. The authors suggest that the any misalignment of the field will destroy the (2:2)x phase, as it resides on “a saddle point in the energy landscape”. It is not obvious to me - what other configurations become lower in energy then? Can one find such instability in the models discussed in the manuscript?

5. Figure 3. What is the meaning of the sentence “The error bars on the extracted Jeff values are uncertainties obtained from the fits.” The values plotted with error bars are the not Jeff...

6. In the discussion of the extracted Jeff value (page 4) the authors claim that the agreement between this value and the prediction from the model including the long range interactions is remarkable. Again, this statement is not completely clear to me. What values are compared here? A model including dipolar interactions is used in Monte Carlo simulations in this work. What is then the energy difference (without the Zeeman term) between the (3:1) and (2:2)x or (2:2)o states? Can it explain the dependencies of critical field vs angle obtained for investigated transitions? What would change if next nearest neighbors exchange interactions were included? I am asking only about the energies per tetrahedron calculated for configurations discussed in this work (not Monte Carlo simulations). If I understand correctly, energy differences between configurations could be used to plot the field vs angle dependence for the investigated transitions.

7. Notation in the expression for the critical field required for the transition to Kagome ice seems to be inconsistent with the rest of the article.

Response to reviewers:

We have revised the manuscript addressing all concerns raised by the reviewers. We would like to thank the reviewers for their insightful comments and questions. Changes in the manuscript are highlighted in red text. Responses to the reviewers are below, indicated in red text.

Reviewer #1 (Remarks to the Author):

This manuscript presents a study by torque magnetometry, augmented by modeling and Monte Carlo simulations, of the high-field phases of a classical spin ice material, $\text{Ho}_2\text{Ti}_2\text{O}_7$. The data shown are of a high quality, and are analyzed and discussed in a compelling way. The established phase diagram will certainly be of interest to the workers in the field, and overall the study clearly deserves dissemination.

However, in my opinion it is less clear whether Nature Communications or a more specialized journal is a more appropriate outlet. In both the abstract and the concluding paragraph, the authors stress the method of torque magnetometry – or more specifically capacitive torque magnetometry (CTM) as having a unique capability in investigating spin ices and other highly anisotropic systems. CTM is not a very new technique, and together with other types (e.g. piezoresistive) torque magnetometry has been long used to study anisotropic systems (such as anisotropic superconductors). With regard to establishing phase boundaries in spin ice materials (and to link those as well as additional magnetic information to model hamiltonians), other magnetic techniques seem to work as well (for example, ac susceptibility combined with a vector magnet, had been applied by Borzi et al., Nat. Commun. 7:12592 on $\text{Ho}_2\text{Ti}_2\text{O}_7$ and $\text{Dy}_2\text{Ti}_2\text{O}_7$ to that effect (although the region of fields and angles probed in that study was more limited). Apart from the methodological aspect, the manuscript also presents conclusions about the physics of $\text{Ho}_2\text{Ti}_2\text{O}_7$. As mentioned in the last sentence of the abstract, the analysis with Monte Carlo simulations suggests that longer-range than nearest neighbor exchange interactions need to be included in the Hamiltonian for spin ices. This is convincingly demonstrated, although the paper could have gained by trying to estimate what those further terms should be to be consistent with the experimental results. From past publications (including the specifically cited refs. 16,21, 22 and the paper by Borzi and coworkers) it seems to me that this was already pretty well established before. In summary, while I do think this is a very nice paper, I do not see the proof of an important advance established by the manuscript, at least in its present form.

We agree with the reviewer in that the torque technique is not new, however, it is traditionally used to measure the magneto-crystalline anisotropy of materials, not to measure the effective spin-pair interaction strength. Additionally, this technique has not been applied to the investigation of spin-ice systems before and we clearly show it can be successfully used to extract information about the interaction parameters that govern the physics in spin ices. We also agree that susceptibility measurements could be used to measure the phase transitions between spin textures in a similar way. Importantly, the accuracy with which we can perform torque measurements, i.e., the precision with which we can align the magnetic field with specific crystallographic directions, is crucial for the success of the experiments. The available rotation range in our experiments allows us to extract the J_{eff} for each of the two spin sublattices, separately. As the reviewer indicated, the measurements in Borzi et al. do not have the angular range needed to extract the J_{eff} for each of the two transitions directly. Furthermore, while they appear to model the susceptibility data for $\text{Dy}_2\text{Ti}_2\text{O}_7$ (DTO), no such modeling is done for the HTO data. Hence, no J_{eff} values are provided for HTO in Borzi et al. and to the best of our knowledge, if values are reported for J_2/J_3 in literature they are for DTO not HTO. What our paper then

shows is that the second and third neighbor couplings are necessary to understand the HTO phase diagram and we extract a value for the overall J_{eff} , which is associated with the internal field that the α and β spin sublattices interact with. From these measurements it is clear that the two sublattices see different internal fields and have different MCPs for monopole nucleation, which has not been reported before. Furthermore, Borzi et al. show that the second and third neighbor couplings are required to describe the low temperature ($T < 0.2$ K) ordered states with alternating single or double spin chains. Our work clearly shows the importance of these couplings for the formation and stabilization of the $(2:2)_x$ phase (alternating spin chains at intermediate temperatures). While our findings are consistent with what Borzi et al., has reported, our measurements indicate that the value for J_{eff} is significantly enhanced when the alternating beta spin chains are involved in the magnetic phase transition. We have added Monte Carlo (MC) simulations using a more generalized DSI model, which includes nearest neighbor exchange (J_1), long-range dipolar (D), and next nearest neighbor (J_2) interactions. We have varied the J_2 term (up to 0.03 meV) and found improved correspondence between the simulated and measured torque curves, especially around the transition that involves the α -spin sublattice (around the [112] direction). We provide an estimate for $J_2 \sim 0.35$ K, which is larger than for DTO. Strikingly, while the introduction of J_2 does lead to increased angular stability of the $(2:2)_x$ phase, J_2 needs to be much larger still to capture the measured angular stability of the $(2:2)_x$ phase, which would compromise the model-data correspondence for the $(2:2)_0 \leftrightarrow (3:1)$ transition. Thus, the introduction of a substantial J_3 term is needed to explain the $(2:2)_x$ phase stability. A full re-optimization of all exchange parameters may be needed since each term can renormalize the others, owing to the strongly correlated nature of the system, which will be part of future work. Furthermore, our work shows that while DTO and HTO are considered similar, J_1 and J_2 are quite different between the compounds. We believe that we have shown that CTM can routinely serve as a natural complement to other techniques, to put stringent bounds on effective Hamiltonians and theories of magnetic materials, thereby aiding a complete understanding of their low-energy properties and response to magnetic fields.

We have added the simulated torque curves with $J_2 = 0.35$ K at $B = 2$ and $B = 11$ T to fig. 4 of the manuscript, simulated curves with smaller J_2 values have been added to the Supplementary Materials (Fig. S6). We have modified the last paragraph of the introduction and added a paragraph in the discussion of the MC simulations, both are copied below,

“A striking result is that the extracted MCP associated with monopole formation is different, depending on whether the monopoles nucleate on the α or β spin sublattices (see Fig.1(c)-(f)). While this conclusion is supported by the standard DSI-model-based classical Monte Carlo simulations (including dipolar D , see Fig. 1(b)), a comparison with the data clearly reveals the shortcomings of this form of the DSI Hamiltonian. Addition of a next-nearest neighbor exchange term ($J_2 \sim 0.35$ K) improves the correspondence for the transition involving the α -spins, providing an estimate for J_2 . This term also helps to stabilize the $(2:2)_x$ state, due the additional Ising antiferromagnetic interaction between the β spin chains, however, the correspondence for this transition is still lacking. Additional third-nearest neighbor exchange terms (J_3^a , and J_3^b) are therefore required to fully describe the field-induced phase transitions in HTO. The idea of needing long-range exchange interactions for the complete description of spin ices is not new. Values for $\text{Dy}_2\text{Ti}_2\text{O}_7$ have been extracted via modeling of susceptibility and neutron data [11, 12, 14, 25], but to the best of the authors’ knowledge, no such modeling has been reported for HTO.”

“To extend the DSI model beyond just the nearest-neighbor and dipolar terms, the minimal way is to add a next nearest-neighbor J_2 interaction. The presence of J_2 does not change the energetics of the

(3:1) phase, but for $J_2 > 0$ (see Methods) an additional Ising antiferromagnetic interaction is introduced. We expect this to lower the energy of the $(2:2)_x$ phase, making it more stable over a wider angular range. We have simulated curves for various J_2 values up to 0.03 meV (~ 0.35 K). In Fig. 4 we plot the torque curve associated with s-DSI with $J_2 \sim 0.35$ K added to it. This term improves the agreement between the data and the MC simulations for the transition involving the α -spins, now accurately approximating the (3:1) stability at low fields, providing an estimate for the size of J_2 for HTO. This value of $|J_2/J_1|$ is similar (but higher than) the reported value for the sister compound $Dy_2Ti_2O_7$ [11, 25]. However, while the angular stability of the $(2:2)_x$ phase did noticeably increase, the quantitative value of the angular extent (see inset) is not fully explained by the J_2 term, indicating that interactions such as J_3 , are necessary for a precise characterization of the Hamiltonian. Owing to the strongly correlated nature of the system, a full re-optimization of all exchange parameters may be needed. Further theoretical studies to confirm this interesting result will be part of future work. We envisage that the sensitivity of the CTM measurements, possibly in combination with other methods such as neutron scattering, specific heat, and bulk magnetization will be instrumental in determining the optimal parameters for $Ho_2Ti_2O_7$."

We believe the revised manuscript is suitable for Nature Communications. We note that reviewer 2 finds the study to be of high quality and states "I am therefore prepared to recommend the manuscript for publication in Nature Communications."

Reviewer #2 (Remarks to the Author):

The manuscript under consideration employs capacitive torque magnetometry technique to explore the magnetic phase diagram of the spin-ice material $Ho_2Ti_2O_7$. The authors focus on the higher-field region where they are able to extract the critical angles from their experimental data. Overall, the manuscript makes a good impression by providing new experimental data on a well-studied compound backed-up by the Monte Carlo simulations. Supplementary materials give detailed information on the geometry of the measurements, field sweeps etc. proving author's very careful approach to the data analysis. I am therefore prepared to recommend the manuscript for publication in Nature Communications. The only comment I have is related to the hysteresis observed in lower fields. It has been shown from the low-temperature magnetisation measurements that field-sweeping rate is important around the spin freezing temperature (the relaxation times become very large). Could it be that the speed of changing the angle for applied field is also important in that respect and that if the direction is changed rather rapidly, the system is effectively out of the equilibrium?

We agree with the reviewer and we have added a statement about this in the supplemental materials.

"The hysteresis is a clear sign of the glassiness of the system. The field rotation speed may be important in determining the degree of hysteresis, as the system will be effectively out of equilibrium if the field rotates too fast."

Further comments: Page 1, the authors claim that "... these systems have been mainly studied by magnetometry, neutron scattering, analytically, or through numerical methods" This is largely true,

however, the dilatometry and ultrasound techniques have also been used for probing field-induced states in the spin ice compound, they should also be mentioned and the appropriate references given.

This has been addressed.

Fig. 2. The x-axis labelling for panel (a) should be made identical to the one used on panel (c), as the marks for 50, 100, 150 degrees are not particularly useful. Fig. 3. A second caption above panel (a) is unnecessary. Same comment for Fig. 4(b). Supplementary materials. Not sure the meaning of 'measurements' in the section 5 title, "Hamiltonian, Monte Carlo Simulations and Measurements".

This has been addressed.

Reviewer #3 (Remarks to the Author):

The manuscript presents a study of transitions between ordered states in spin ice compound $\text{Ho}_2\text{Ti}_2\text{O}_7$ under external magnetic field rotating in two planes. The capacitive torque magnetometry is used to detect spin flip transitions as a function of the strength of the external field and the field direction. A model including nearest neighbor interaction and the Zeeman term is fitted to the experimental data and the effective nearest neighbor interaction constants are obtained. In my opinion the manuscript is in many places unclear or inconsistent and it is difficult to assess the main results and the novelty of the study. Some additional explanations would be needed for the clarity of the paper, and there are also points that I do not understand. For example, the estimation of the effective interaction term J_{eff} is one of the key results of the study, however, J_{eff} is not properly defined or introduced at the beginning of the paper and it is used in a way that I find confusing. Several detailed comments are listed below.

1. I assume J_{eff} to be the effective nearest neighbor interaction between spins, defined as in the caption of Fig. 1 and leading to energy levels shown in the left panel of Fig 1a. However, I do not understand the notation in the right panel of Fig 1a (not adequately explained in the caption):- the axis label "Ez": E_z is defined later in the text as the Zeeman energy, however, there is also the interaction term leading to the energy levels plotted in the figure, - for B [100], the splitting of the energy levels of (2:2) states should be 1:4:1, not 1:5. Also, the (3:1) states would split. The energy difference 3.2 K, if I understand correctly, comes from the spin-spin interaction term, so the gap between the lowest (2:2) state and the lowest (3:1) state should be modified,- for B[110], the authors decided to show levels of only two (2:2) states that are the building blocks of the (2:2)x configuration with larger unit cell. However, the (3:1) states should be split, if all 8 levels are shown.- the $B > 2$ T condition should be clarified.

We thank the reviewer for this comment. Panel (a) (right side) of the figure has been removed. We now show the energies for one tetrahedron for each of the possible spin textures at zero field, these energies are based on $J_{1,\text{eff}}$ (nearest neighbor exchange interactions only). Panel (b) of the updated figure provides an overview of how the effective spin-pair interaction changes as dipolar interactions truncated at nearest neighbor or long-range dipolar interactions are taken into account. The generalized dipolar spin ice model (Borzi et al) takes into account second and third nearest neighbor exchange interactions, which is expected to lead to changed effective spin-pair interactions. While values for these next and next-next nearest neighbor interactions have been determined for DTO, no values have been

reported for HTO. From our measurements, we extract the internal field (i.e., the effective spin-pair interaction strength) that the field-decoupled spins experience with the rest of the spin lattice. We find that the internal field is not the same for the α and β sublattices (depicted in the updated panel c of Fig. 1). In the revised version we provide an estimate of J_2 for HTO, which is reported in the table, investigation of J_3 values will be part of a future study. CTM can routinely serve as a natural complement to other techniques, as it can put stringent bounds on effective Hamiltonians and theories of magnetic materials.

2. J_{eff} is described in the text as field dependent, or field-directiondependent. After reading the whole text I suppose it means that (two) different values of J_{eff} were extracted from the data, depending on the magnetic field direction during the transition from one of (2:2) states to a (3:1) state. As discussed in the text, the different values might result from long range interactions present in the spin states involved in these transitions. Describing J_{eff} as field-dependent might be confusing.

We agree that the wording was confusing, we have updated the manuscript to clarify that we are extracting two internal fields and effective spin-pair interactions strengths depending on the field direction, the spin-sublattice involved, and the ordered spin ice states that participate in the transitions.

3. A term “field decoupled spins” is used several times in the manuscript. Indeed, one or two spin sublattices become perpendicular to the field for some direction of B, for example B parallel to [112] or [110]. However, I do not understand some statements in the text:- J_{eff} described as the interaction strength between field-decoupled spins (page 1),- in Figure 1, the light blue box is described as indicating the spin sublattice that becomes field decoupled when the field is “tilted away from the [112] direction” (the spins are decoupled from field precisely for [112] direction, and become coupled to a field component as the field changes),- “By identifying the field-decoupled spin sublattice for each transition, we fit the extracted angles against critical field and extract the MCP associated with (3:1) monopole creation/annihilation” [when B is rotated, the Zeeman energy of a given spin sublattice changes and this might lead to spin flips in this sublattice. However, not always spins decoupled for some field direction are flipped in the transition, for example only half of the beta spins, decoupled at B parallel to [110], flip in the transition from a (2:2) to (3:1) state].

We agree that the wording was confusing, we have updated the manuscript to clarify. The sentence on page 1 was changed, it now reads, “In this work, we employ CTM measurements to characterize the spin-ice system $\text{Ho}_2\text{Ti}_2\text{O}_7$ (HTO) and to measure the effective spin-pair correlation strength between field-decoupled spins and the mean field.”

We also added the following sentence to the introduction:

“The effective spin-pair coupling (J_{eff}) determines the energy per tetrahedron for each of these states; for nearest neighbor exchange interactions, $2J_{1,\text{eff}}$ is required to trigger the (2:2) \rightarrow (3:1) transition. Importantly, the value of J_{eff} is altered if interactions beyond nearest neighbor (i.e., dipolar D and 2nd and 3rd nearest neighbor exchange J_2 , J_3) are included (Fig. 1 (a) and (b)), as described in previously reported model [1,11-14].”

We have updated the wording in the methods sections as well. The passage indicated by the reviewer now reads,

“While one may expect that the α and β spin sublattices decouple exactly at $\langle 1\ 1\ 2 \rangle$ and $\langle 1\ 1\ 0 \rangle$ field directions, respectively, the internal field produced by mean field will shift that transition to a critical angle away from these crystallographic directions. Thus, the Zeeman energy (E_z) associated with this critical angle is a direct measure of this internal field. We calculate the analytic form of the Zeeman energy of the field-decoupled spins as the field rotates past the $\langle 1\ 1\ 2 \rangle$ and $\langle 1\ 1\ 0 \rangle$ crystallographic directions. Expressing E_z in terms of applied field (B) and field direction (θ), and realizing that $E_z=2J_{\text{eff}}$, allows us to determine a fitting function for the field vs. critical angle data from which the change in MCP (J_{α}^{eff} and J_{β}^{eff}) associated with the proliferation of (3:1) tetrahedra can be determined”

4. The authors suggest that the any misalignment of the field will destroy the (2:2)_x phase, as it resides on “a saddle point in the energy landscape”. It is not obvious to me - what other configurations become lower in energy then? Can one find such instability in the models discussed in the manuscript?

We appreciate the comment of the reviewer, and acknowledge that this was not clearly explained. The key point here is that when the field is confined to rotate in the xy-plane there is no sign of an ordered phase associated with the [110] direction. Hence, the ordered alternating chains do not form when the field rotates in this plane, in stark contrast with the (2:2)_x phase that is clearly visible in the (1-10) rotation plane. The state that forms around the [110] in the xy-plane rotation is a domain state of 2in/2-out textures on each tetrahedron with net moments in the x and y directions, but no order on the β spin sublattice. We have updated the manuscript to clarify this point.

“To visualize the surprising anisotropy in (2:2)_x stability between the two rotation planes, is to explore the energy surface that is obtained by integrating the torque curves. We show the energy surface contours associated with the (0 0 1) and the (1-1 0) rotation planes in the Supplementary Materials[26], with (2:2)_x residing on a sharp maximum in the (0 0 1) plane and on a local minimum in the (1-1 0) plane. Thus, (2:2)_x resides on a saddle point in the energy landscape. While it is quite robust against misalignment of the field in the in the (1-10) plane, in the (001) plane the (2:2)_x is not stable, and the system favors the (2:2)₀ states (with $m \parallel \langle 100 \rangle$).”

The following figure and an accompanying description was added to the Supplementary Materials.

Fig. S3: Field scaled integrated torque curves based on the curves shown in Fig. 2 of the main manuscript. These curves represent contours on the magnetic anisotropy energy surface as a function of angle. a) the sample is rotated so that the field stays in the (001) plane and b) in the (1-10) plane.

5. Figure 3. What is the meaning of the sentence “The error bars on the extracted J_{eff} values are uncertainties obtained from the fits.” The values plotted with error bars are the not J_{eff} ..

The plotted error bars associated with the data points in Fig.3 are a visualization of the residuals. The uncertainty of the J_{eff} value, which is determined from fitting the B vs. angle plots using the derived analytical expressions (see Methods), comes from the error generated by the fitting procedure. We added the following sentence to the caption “The plotted error bars on the data points are a visualization of the residuals of the fits.”

6. In the discussion of the extracted J_{eff} value (page 4) the authors claim that the agreement between this value and the prediction from the model including the long range interactions is remarkable. Again, this statement is not completely clear to me. What values are compared here? A model including dipolar interactions is used in Monte Carlo simulations in this work. What is then the energy difference (without the Zeeman term) between the (3:1) and (2:2)_x or (2:2)_o states? Can it explain the dependencies of critical field vs angle obtained for investigated transitions? What would change if next nearest neighbors exchange interactions were included? I am asking only about the energies per tetrahedron calculated for configurations discussed in this work (not Monte Carlo simulations). If I understand correctly, energy differences between configurations could be used to plot the field vs angle dependence for the investigated transitions.

The energies per tetrahedron for each of the plotted states in zero field are shown in Fig. 1a). The energy differences are set by the effective spin-pair interaction strength, i.e., the internal field that each of the spins “sees” from the rest of the spin lattice. While these energy differences are expressed as $J_{1,\text{eff}}$, i.e., based on the nearest neighbor exchange interaction only, in panel a), the effective spin correlation strength will vary as dipolar and higher order exchange interaction terms are taken into account. We provide an overview of how J_{eff} changes for the different reported models in panel b) of Fig. 1. Thus, Fig. 1 b) shows how the energy differences between the different states changes when long-range interactions are introduced. Note, values for J_2 and J_3 have not been determined previously for HTO, but in the revised manuscript we provide an estimate for J_2 . As stated above, integrated torque curves represent the magnetic anisotropy energy. Thus, the calculated torque curves in Fig. 2 are directly related to these energy differences. By measuring the critical angles associated with the magnetic phase transitions we directly measure the internal field and thus we find a value for J_{eff} , which depends on the spin-sublattice and the ordered spin ice states that are involved in the transitions. We are comparing our measured values extracted from actual torque measurements ($J_{\text{eff}}^{\alpha} = 1.61(5)$ K and $J_{\text{eff}}^{\beta} = 2.2(1)$ K) to the values reported on by others (Fig. 1b)) and to the values extracted from the Monte Carlo simulations ($J_{\text{eff}}^{\alpha,\text{MC}} = 1.4(2)$ K and $J_{\text{eff}}^{\beta,\text{MC}} = 1.8(1)$ K). This is more clearly indicated in the updated manuscript.

We have added MC simulations using a more generalized DSI model, which includes nearest neighbor exchange (J_1), long-range dipolar (D), and next nearest neighbor (J_2) interactions. We have varied the J_2 term (up to 0.03 meV) and found improved correspondence between the simulated and measured torque curves, especially around the transition that involves the α -spin sublattice (around the [112] direction). We provide an estimate for $J_2 \sim 0.35$ K, which is larger than for DTO. Strikingly, while the

introduction of J_2 does lead to increased angular stability of the $(2:2)_x$ phase, J_2 needs to be much larger still to capture the measured stability, which would compromise the model-data correspondence for the $(2:2)_0 \leftrightarrow (3:1)$ transition. Thus, it seems that introduction of a substantial J_3 term is needed to explain the $(2:2)_x$ phase stability. Additionally, a full re-optimization of all exchange parameters may be needed since each term can renormalize the others, owing to the strongly correlated nature of the system, which will be part of future work.

7. Notation in the expression for the critical field required for the transition to Kagome ice seems to be inconsistent with the rest of the article

We thank the reviewer for pointing this out, we have addressed this issue in the updated manuscript.

REVIEWERS' COMMENTS

Reviewer #1 (Remarks to the Author):

The present version of the manuscript is provided with additional simulations, extracting a value for the next-nearest-neighbor exchange coupling J_2 . This improves the agreement between experimental data and modeling, although as the authors state, remaining discrepancies point to the need of incorporating at least one further exchange coupling parameter – which is left for future work. This goes some ways to achieving the same level of detail modeling as for $\text{Dy}_2\text{Ti}_2\text{O}_7$, allowing comparing differences in the model parameters and their effects.

Concerning the CTM method, in the revision the authors have strengthened their case for its usefulness as a complementary tool in the study of this material and spin ices in general.

Both of the above, in my view, add to the significance of the manuscript – and together with the finding of different J_{eff} for the two spin sublattices already pointed out in the previous version of the manuscript, the threshold for this journal may be met.

Reviewer #2 (Remarks to the Author):

My initial assessment of the manuscript in question was generally positive, I have only indicated some minor points to be considered by the authors.

In the revised version, the authors have addressed all of my concerns, I'm therefore happy to recommend the manuscript for publication.

Reviewer #3 (Remarks to the Author):

In the corrected manuscript all the points from my review have been addressed. Figure 1 is much more understandable and some other inconsistencies in the notation have been removed.

I am still confused by a statement in the caption of Fig. 1: “the light blue boxes in (d) and (f) indicate the spin sublattice that becomes decoupled from the field when a magnetic field is tilted away from the [1 1 2] direction.” If I understand correctly, this sublattice becomes coupled to the magnetic field as it deviates from from that direction. Except for this sentence, the interplay between the external magnetic field and internal interactions has been clarified.

I am still wondering (point 6 of my review) if some insight can be achieved by looking at the energy of different ordered states and various model parameters, without doing Monte Carlo simulations. However, the authors explained thoroughly what has been calculated, and additional results are included that give estimation of the next nearest neighbor interactions.

I can now recommend the manuscript for publication in Nature Communications.

We have revised the manuscript addressing all concerns raised by the reviewers. We would like to thank the reviewers for their insightful comments and questions. We believe that our revised manuscript is now ready for publication in Nature communications. Changes in the manuscript are highlighted in red text. Responses to the reviewers are below indicated in red text.

Reviewer #1 (Remarks to the Author):

The present version of the manuscript is provided with additional simulations, extracting a value for the next-nearest-neighbor exchange coupling J_2 . This improves the agreement between experimental data and modeling, although as the authors state, remaining discrepancies point to the need of incorporating at least one further exchange coupling parameter – which is left for future work. This goes some ways to achieving the same level of detail modeling as for $\text{Dy}_2\text{Ti}_2\text{O}_7$, allowing comparing differences in the model parameters and their effects.

Concerning the CTM method, in the revision the authors have strengthened their case for its usefulness as a complementary tool in the study of this material and spin ices in general.

Both of the above, in my view, add to the significance of the manuscript – and together with the finding of different J for the two spin sublattices already pointed out in the previous version of the manuscript, the threshold for this journal may be met.

We appreciate the valuable input and positive assessment provided by the reviewer

Reviewer #2 (Remarks to the Author):

My initial assessment of the manuscript in question was generally positive, I have only indicated some minor points to be considered by the authors.

In the revised version, the authors have addressed all of my concerns, I'm therefore happy to recommend the manuscript for publication.

We appreciate the valuable input and positive assessment provided by the reviewer

Reviewer #3 (Remarks to the Author):

In the corrected manuscript all the points from my review have been addressed. Figure 1 is much more understandable and some other inconsistencies in the notation have been removed.

I am still confused by a statement in the caption of Fig. 1: “the light blue boxes in (d) and (f) indicate the spin sublattice that becomes decoupled from the field when a magnetic field is tilted away from the [1 1 2] direction.” If I understand correctly, this sublattice becomes coupled to the magnetic field as it deviates from from that direction. Except for this sentence, the interplay between the external magnetic field and internal interactions has been clarified.

We appreciate this comment from the reviewer. The reviewer is correct, the highlighted spins become decoupled from the field when the field points exactly along the [112] direction. The coupling to the external field, once it is rotated beyond the [112], provides the Zeeman energy necessary to induce spin reversal of these spins. The critical angle at which reversal occurs, corresponds to the internal field imposed by the other spins in the lattice (i.e., it is a measure of the spin-correlation strength). As pointed out by the reviewer, this point was already clarified in the main text. We have updated the sentence in the caption, it now reads,

“The spins denoted by the light blue boxes in panels (a) and (c) indicate the spin sublattice that becomes decoupled from the field when the magnetic field is directed exactly along the [112] direction.”

I am still wondering (point 6 of my review) if some insight can be achieved by looking at the energy of different ordered states and various model parameters, without doing Monte Carlo simulations. However, the authors explained thoroughly what has been calculated, and additional results are included that give estimation of the next nearest neighbor interactions.

We appreciate this comment by the reviewer and realize that we had not sufficiently answered this question in our previous response. The point of the reviewer, that the energy of the ordered states can be determined (without MC simulations) is correct. Note, while energies for single tetrahedra can be determined in a strictly nearest neighbor model (considering J_1 only), one has to go to 128 sites (8 unit cells) to capture the effect of J_2 and J_3 interactions. We have calculated the energies of the ordered states by enumerating all the 1st, 2nd and 3rd nearest neighbors. We computed the interaction energy and the Zeeman energy per 16-site unit cell for the different phases. By equating the total energies (interaction + Zeeman) for two different phases, one can determine at what angle the transition would occur if the values for interaction parameters (J_1 , J_2 , J_{3a} , J_{3b} and D) are known.

This analysis provides an easy way to see that a J_2 term only affects the transition between the $(2:2)_0$ and $(3:1)$ phases, while any J_3 -terms only affects the $(2:2)_x$ and $(3:1)$ transitions. Hence, we had already extracted J_2 , we can now also provide a rough estimate of the value of $(J_{3a} + J_{3b})$ for HTO using our CTM experiments.

We have described all of this in the Supplementary Information Note 7 and we have added a paragraph discussing this point in the main text. It reads,

“We support our findings with a short-range phenomenological model, which we use to evaluate the interaction energy for each spin-ice phase (see Supplementary Note 7 for more details). From this analysis, one can see what effect each of the interaction terms in the Hamiltonian has on the phase boundary of the field-induced magnetic phase transitions in HTO. In short, for the $(2:2)_0 \leftrightarrow (3:1)$ transitions, the introduction of a J_2 -term affects the interaction energy of the $(2:2)_0$ state, but does not impact the energetics of the $(3:1)$ state. Effectively, J_2 partially negates the effects of long-range dipolar interactions. Note, adding a J_3 -term affects both the $(2:2)_0$ and $(3:1)$ states in the same way, thus this effect cancels out when evaluating the location of the phase boundary associated with this transition. (These J_3 terms correspond to two different kinds of third nearest neighbors, their couplings are referred to as J_{3a} and J_{3b} , see Supplemental Note 7). For the $(2:2)_x \leftrightarrow (3:1)$ transitions, the introduction of J_2 also does not affect the energetics of the $(2:2)_x$ phase, as the interaction energy associated with this term sums to zero (i.e., similar to the $(3:1)$ phase). Hence, the phase boundaries of the $(2:2)_x \leftrightarrow (3:1)$

transitions are unaffected by the J_2 term, a finding broadly consistent with the MC simulations. However, the J_3 terms affect the $(2:2)_x$ and $(3:1)$ phases differently, and are therefore important in determining the location of the phase boundary for this transition. Thus, this simple short-range model allows us to constrain the value for $J_{3a} + J_{3b}$ to a ball-park value of ~ -0.014 meV (-0.16 K). The sign and the order of magnitude for $J_{3a} + J_{3b}$ are consistent with previously reported values for DTO [11].”

I can now recommend the manuscript for publication in Nature Communications.

We appreciate the valuable input and positive assessment provided by the reviewer

Additionally, in the earlier manuscript we referred to crystallographic data that was not shown (Methods section). We have fixed that issue and deposited the data in a repository, which we now cite, and we have added some details about the crystallographic measurements in the Supplementary Information (Note 8). In the main text we added the following, “Single crystal x-ray diffraction experiments, taken on an Oxford-Diffraction Xcalibur-2 CCD diffractometer equipped with a graphite-monochromated $\text{MoK}\alpha$ source, confirm the symmetry ($Fd-3m$) and lattice parameter of $10.0839(1)$ Å at 293 K, consistent with previous reports [3] (see Supplementary Note 8 for more details).”

The energies in Fig.1 a) were altered by a factor of 2 and we put the $(3:1)$ state at zero energy. Note, the previous figure had the $(2:2)$ states at $-2J_{\text{eff}}$, this is correct for an isolated tetrahedron, however for a spin in the pyrochlore lattice, nearest neighbors reside in two (one “up” and one “down”) tetrahedra. Thus, taking into account all nearest neighbors, an additional factor of 2 is obtained for the energy of the states in a pyrochlore lattice. This notation is more consistent with panel b) of the same figure.

All the above-mentioned changes and other minor changes in the main manuscript are highlighted in red text. These include sentences that have been slightly reworded and/or that have been moved.